# Midpalatal Suture Density Evaluation after Rapid and Slow Maxillary Expansion with a Low-Dose CT Protocol: A Retrospective Study

**DOI:** 10.3390/medicina56030112

**Published:** 2020-03-05

**Authors:** Rosamaria Fastuca, Ambra Michelotti, Riccardo Nucera, Vincenzo D’Antò, Angela Militi, Antonino Logiudice, Alberto Caprioglio, Marco Portelli

**Affiliations:** 1Department of Medicine and Surgery, School of Medicine, University of Insubria, Via G. Piatti 10, 21100 Varese, Italy; rosamariaf@hotmail.it (R.F.); alberto.caprioglio@uninsubria.it (A.C.); 2Section of Orthodontics, Department of Neuroscience, Reproductive Sciences and Oral Sciences, University of Naples Federico II, Via Pansini, 5, 80131 Naples, Italy; ambramichelotti@gmail.com; 3Department of Biomedical Sciences, Dentistry and Morphological and Functional Imaging, University of Messina, Via Consolare Valeria 1, 98100, Messina, Italy; riccardon@unime.it (R.N.); amiliti@unime.it (A.M.); nino.logiudice@gmail.com (A.L.); 4School of Orthodontics, University of Naples, 80131 Naples, Italy; vincenzo.danto@unina.it

**Keywords:** maxillary expansion, bone density, computed tomography

## Abstract

*Background and objectives:* The aim of the present paper is to use low-dose computed tomography (CT) to evaluate the changes in the midpalatal suture density in patients treated with rapid maxillary expansion (RME) and slow maxillary expansion (SME). *Materials and Methods:* Thirty patients (mean age 10.2 ± 1.2 years) were retrospectively selected from the existing sample of a previous study. For each patient, a low-dose computed tomography examination was performed before appliance placement (T0) and at the end of retention (T1), seven months later. Using the collected images, the midpalatal suture density was evaluated in six regions of interest. *Results:* No significant differences were found between the timepoints in the rapid maxillary expansion group. Three out of six regions of interest showed significant decreases between the timepoints in the slow maxillary expansion group. No significant differences were found in comparisons between the two groups. *Conclusions:* The midpalatal suture density showed no significant differences when rapid maxillary expansion groups were compared to slow maxillary expansion groups, suggesting that a similar rate of suture reorganization occurs despite different expansion protocols.

## 1. Introduction

Maxillary expansion is a widely used orthodontic technique that enables the correction of transversal upper-arch deficiency [1,2], a condition which can be related to both genetic and environmental factors [3]. Genetic factors [4] account for narrow developed maxilla and/or wide mandible and maxillary deficiencies in cases of cleft lip and palate patients [5,6]. Environmental factors are related to mouth breathing, which is often associated with posterior nasal obstruction and oral habits [7,8]. Most transversal maxillary discrepancies clinically show up with monolateral or bilateral posterior crossbite. Different treatment possibilities have been investigated to correct maxillary deficiency through the use of a force along the midpalatal suture. Rapid maxillary expansion (RME) treatment is the treatment most commonly used by clinicians [9], though slow maxillary expansion (SME) is being employed more often [10]. The effects of palatal expansion are mainly skeletal and related to the upper arch transverse dimensions, however dental and respiratory function changes are often present [11,12,13,14,15,16,17]. According to a recent systematic review and meta-analysis, the current evidence on the dental and skeletal effects was reported to be of a higher level for RME than SME, but the two expansion protocols seemed to have no significant differences in dentoalveolar transversal effects [17]. However, it is necessary to apply a fixed retainer after the expansion to avoid dental relapses [18]. A good understanding of the midpalatal suture modification after both RME and SME is important in order to evaluate the correct retention timing and avoid a relapse of the skeletal effects induced by palatal expansion. Currently, there is a lack of sufficient evidence to establish which protocol of palatal expansion, rapid or slow, is preferable; for this reason, clinicians often choose the expansion protocol solely on the basis of their personal experience. Studies on animals and human beings have been attempted with the aim of evaluating the effects of expansion procedures on the midpalatal suture. Different studies have found that SME allows greater physiologic modification of the maxillary bone and surrounding complex and prevents the accumulation of large residual loads within a more stable bone remodeling [19]. It is widely known that the RME protocol produces an accumulation of orthopedic forces in the maxillary bone and in the surrounding structures that slowly dissipates; this condition can affect the bone remodeling processes. Radiographic examinations, such as posteroanterior cephalograms [20] and occlusal radiographs [21], have been employed to study the dimensional changes in the midpalatal suture produced by RME in growing subjects. In other studies [22,23,24], the midpalatal suture density after RME was evaluated with low-dose computed tomography (CT) using the Hounsfield index at different stages, but no conclusive evidence was detected about the midpalatal suture changes after treatment and retention [25]. The aim of this study is to evaluate the midpalatal suture density in growing patients who have undergone RME and SME using low-dose CT, in order to evaluate the skeletal effects related to these different protocols of palatal expansion.

## 2. Materials and Methods

This study followed a retrospective design. Patients were selected from the existing samples of previous studies [10,15] of subjects needing orthodontic treatment who had not been treated before and who attended the Section of Orthodontics, Department of Medicine and Surgery, University of Insubria, Varese, Italy; the Section of Orthodontics, Department of Oral Sciences, University of Naples Federico II, Italy; or the Section of Orthodontics, Department of Biomedical, Dental and Functional Images Science, University of Messina, Italy. Signed informed consent for releasing these diagnostic records for scientific purposes was acquired from the patients’ parents prior to the beginning of the treatment. The protocol was reviewed and approved by the Ethical Committee, (Approval n° 826, 7 February 2005) and the procedures followed adhered to the World Medical Association’s Declaration of Helsinki. The inclusion criteria were as follows: (1) good general health, according to medical history and clinical judgment; (2) maxillary transverse deficiency, with or without the presence of unilateral or bilateral posterior crossbite; (3) early mixed dentition; (4) fully erupted upper and lower first molars. Thirty patients (12 females and 18 males with a mean age 10.2 ± 1.2 years) were selected for the study. Fifteen patients (mean age 9.8 ± 0.7 years) had a rapid maxillary expansion protocol (this was the RME group): two bilateral and 13 monolateral crossbites. Fifteen patients (mean age 10.1 ± 0.5 years) had a slow maxillary expansion protocol (the SME group): one bilateral and 14 monolateral crossbites. For each patient, CT examinations were available before appliance placement (T0) and at the end of retention period (T1), seven months later, when the expander was removed. A two-band maxillary expander (TBME) with an 11 mm screw was used for all the subjects (Leone Orthodontic Products, Sesto Fiorentino, Firenze, Italy) (Figure 1).

In both groups, the screw was initially turned eight times (with a 1.60 mm initial transversal activation). Afterwards, patients in the RME group were instructed to turn the screw three times during each subsequent day (0.60 mm activation per day). In the SME group, patients were instructed to turn the screw twice a week (0.40 mm activation per week). The maxillary expansion was performed until dental overcorrection was achieved, defined as when the lingual cuspids of the upper first molars occluded into the buccal cuspids of the lower first molars. After the activation phase, the expander was maintained as a passive retainer. During this period, none of the patients underwent any further orthodontic treatment. The mean active expansion period was 12.9 ± 2.2 days for the RME group and 147.7 ± 7.4 days for the SME group. CT exams have been acquired by the same well trained radiologist using the same CT scanner (MX 8000 IDT6, Philips Medical Systems, Best, The Netherlands). A low-dose CT protocol was used for the acquisitions (KV 80, mAs 28 [26]). Image analysis was carried out using Mimics software, version 10.11 (Materialise Medical Co, Leuven, Belgium). In order to obtain comparable scans along the timepoints, the original head position for each patient was reoriented in a reproducible manner on the basis of defined landmarks (Table 1).

The axial scan passing through hard palate was then identified (Figure 2) and used for all the measurements.

Six regions of interest (ROI) have been traced by one trained operator or the calculation of values of density in Hounsfield units (HU) using the software tools. Four round-shaped ROIs (Figure 3) and two rectangular-shaped ROIs (Figure 4) noted by previous studies [21,23] have been identified as follows.

### 2.1. Round-Shaped ROIs

Anterior suture (AS) ROI: values of density measured in the ROI located along the midpalatal suture 5 mm in front of the center of the nasopalatine duct.

Posterior suture (PS) ROI: values of density measured in the ROI located along the midpalatal suture 5 mm posterior to the center of the nasopalatine duct.

Anterior bone (AB) ROI: values of density measured in the ROI located in the maxillary bone 3 mm laterally to AS ROI (on the right side).

Posterior bone (PB) ROI: values of density measured in the ROI located in the maxillary bone 3 mm laterally to PS ROI (on the right side).

### 2.2. Rectangular-Shaped ROIs

For the identification of the rectangular-shaped ROIs, a rectangular area was selected along the midpalatal suture starting from the landmark located 5 mm in front of the center of the nasopalatine duct and extending the entire length of the suture to the posterior nasal spine (PNS). The width of the rectangle was defined as 3 mm to the left and right side (total width 6 mm) of the center of the nasopalatine duct.

Anterior suture density (ASD) ROI: values of density measured in the ROI located along the midpalatal suture in a rectangular area starting from 5 mm in front of the center of the nasopalatine duct to the anterior half-length of the suture.

Posterior suture density (PSD) ROI: values of density measured in the ROI located along the midpalatal suture in a rectangular area extending from the posterior end of the ASD ROI to posterior half-length of the suture.

All the measurements were performed at T0 and T1.

The sample size was calculated using the measurements of three patients per group, selecting as the main outcome the ASD ROI changes before and after treatment. A sample size of at least 11 subjects was necessary to detect a power of 0.8. Ten randomly selected CT images were retraced by the same operator. Systematic and random errors were calculated, comparing the first and second measurements with dependent *t*-tests and Dahlberg’s formula [27], at a significance level of *p* < 0.05. All measurement error coefficients were found to be adequate for the appropriate reproducibility of the study. A range from 40.32 to 93.75 for density measurements was found. The SPSS software, version 22.0 (SPSS^®^ Inc., Chicago, IL, USA) was used to perform the statistical analyses. All data were preliminary tested for normality and for equal variance through the Shapiro–Wilk test and Levene test, respectively, revealing a normal distribution. Then, parametric tests were employed. Dependent *t*-tests were used to compare the measurements of the RME group and the SME group at the starting forms, and to compare changes due to expansion between timepoints within the same group. An independent *t*-test was used to evaluate the differences between the groups and to compare the sutural ROIs (AS ROI and PS ROI) with the correspondent bony ROIs (AB ROI and PB ROI) at T0. For all the tests, a significance level of *p* < 0.05 was set.

## 3. Results

No significant differences between the two groups were detected in the comparison of the sutural ROIs at the starting forms. Descriptive statistics for the comparison of the timepoints within the same group and the comparison of the changes between groups are shown in Table 2 and Table 3. No significant differences in the sutural ROIs (AS ROI and PS ROI) were detected when compared to the bony ROIs (AB ROI and PB ROI) at T0 in both groups. No significant differences were found between the timepoints in the RME group (Table 2).

PS ROI, ASD ROI and PSD ROI showed significant decreases between the timepoints in the SME group (Table 3).

No significant differences in ROI density were found in the comparison between the two groups (Table 4).

## 4. Discussion

In the present study, the midpalatal suture density changes have been evaluated with low-dose CT images in patients treated with RME and SME before active expansion (T0) and after seven months (T1). All the ROIs were identified on the same axial scan passing through the hard palate. The radiographic density of the maxillary suture was measured along the suture length in the anterior and posterior portions before and after two expansion protocols. Little evidence was reported [22,23,24] on the differences in the sutural density when compared to the maxillary bone density in growing patients. Before treatment, contrasting results are suggested: Franchi et al. found significantly decreased density along the midpalatal suture compared to bone, while Schauseil et al. [22] did not detect any significant difference, in accordance with the present study. These results suggest great variability in the suture morphology and density in growing patients, which is related to age and skeletal maturity. The midpalatal suture density has been evaluated both before and seven months after palatal expansion in order to evaluate the effects of the maxillary expansion treatment on the radiological suture density. The comparative evaluation of the suture density between the RME and SME treatments is important in order to better comprehend the skeletal effects induced by these different protocols and to establish whether a longer period of retention is needed to prevent relapses. If we do not adequately understand the effects of slow and rapid palatal expansion on the suture density, we cannot establish whether retention timing is sufficient to avoid a skeletal relapse of palatal expansion [28]. The RME treatment did not significantly affect the suture density in the present study, and this result is in agreement with a previous investigation [24]. On the contrary, Schauseil et al. [22] reported a significant decrease in the midpalatal suture density after RME. The age of the sample in the study by Schauseil et al. was older than the present study, suggesting that slower regeneration processes might occur within the suture when older patients undergo RME, due to different suture maturation at different ages. When the SME group was analyzed, the suture density showed significant decreases in both the anterior (ASD ROI) and posterior portions (PS ROI, PSD ROI). It is still controversial whether SME might have orthopedic effects on the separation of the midpalatal suture that are similar to RME. High-level evidence on the skeletal effects of SME were not reported [17]. Several studies in animal models [17] have shown that, during expansion, the sutural integrity can be maintained by bone growth. Mossaz and Mossaz [29] evaluated X-ray occlusal films in patients who had undergone SME and noted that no radiolucency could be demonstrated radiographically along the suture, as generally observed in RME. Despite this, there would appear to be additional bone deposition within the suture. Conversely, other studies have demonstrated radiological [30,31] and histological [32] separation of the suture following a slow expansion protocol. The significant decrease in density found in the present study suggests sutural reorganization, but, according to the present methods and results, no evidence was recorded of the presence of sutural separation instead of additional bone deposition within the suture. This result might also be explained with regard to the observation-time interval. The T1 low-dose CT was performed seven months after the beginning of treatment, but since the treatment-time duration was different in the two groups, due to the expansion protocol, the subjects performing the SME underwent a noticeably shorter retention period when compared to the RME group. This might be considered as a limitation of the present study. The comparisons between the RME and SME groups showed no significant differences. Even though significant differences were detected between the timepoints after the SME treatment, the reported changes were not significant when compared to the RME treatment. This result might be explained by the greater variability of the results within the present sample. Slow protocols of expansion showed no significant differences in the skeletal rate of palatal expansion and stability in long-term follow-up, when compared to RME [33], and, according to the results of the present study, no significant differences on the midpalatal suture density. Moreover, a slower expansion rate would prevent the accumulation of large residual forces within the maxillary complex, which may result in the decrease in undesirable concomitant changes in the craniofacial sutures observed during RME [34] which can affect the whole cranio-facial region [35,36,37]. Rapid palatal expansion is also beneficial in cases of class II malocclusion related to a mandibular posterior position induced by a reduced transversal dimension of the maxillary bone [38]. A low-dose protocol of spiral CT was demonstrated to be efficient for the investigation of the maxillo-facial region bone structure [39,40,41,42,43,44,45,46,47], without a significant increase in radiological risk for the patient.

## 5. Conclusions

According to the present results, it is possible to state the following:Bone density measured on the hard palate of prepubertal subjects did not show significant differences in values when compared with the density at the midpalatal suture before treatment;midpalatal suture density showed no significant changes when RME was performed;significant decreases in density were reported after the SME treatment in the whole suture area in the considered time interval, but no significant differences were detected between groups, suggesting similar rates of suture reorganization in spite of differing retention periods.

## Figures and Tables

**Figure 1 medicina-56-00112-f001:**
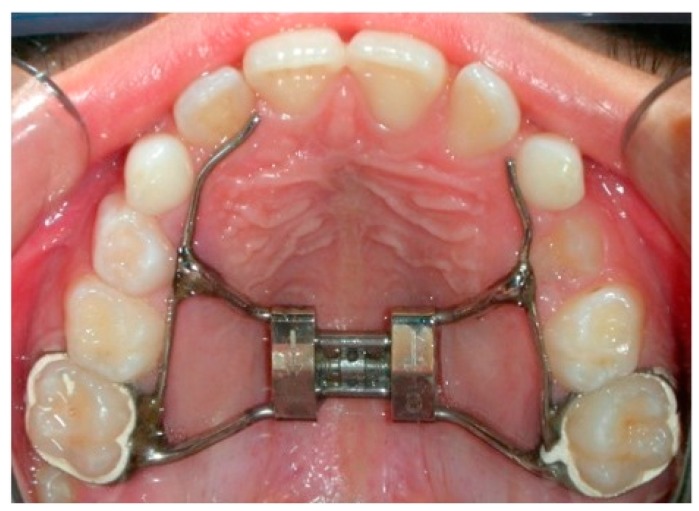
Two-band maxillary expander (TBME).

**Figure 2 medicina-56-00112-f002:**
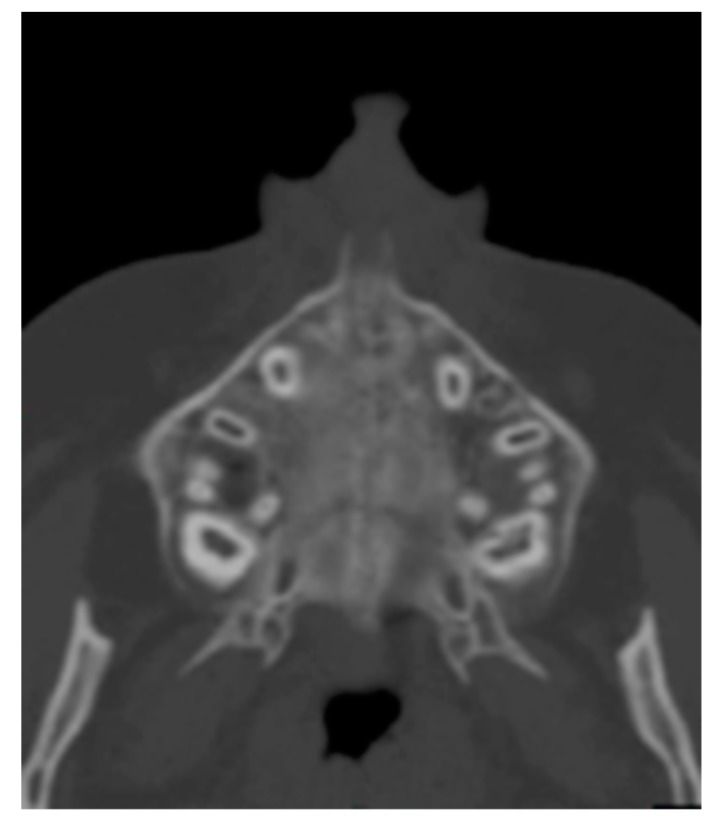
Axial scan.

**Figure 3 medicina-56-00112-f003:**
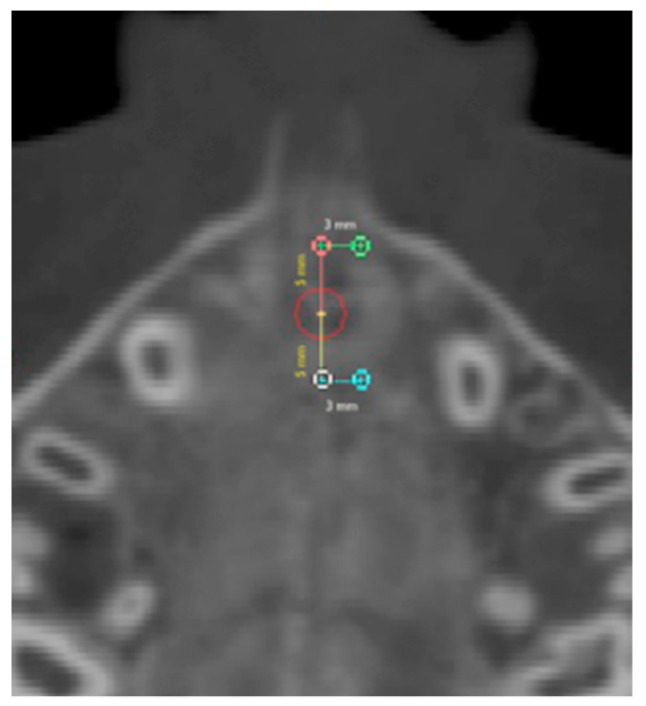
Round-shaped ROIs in the palatal region.

**Figure 4 medicina-56-00112-f004:**
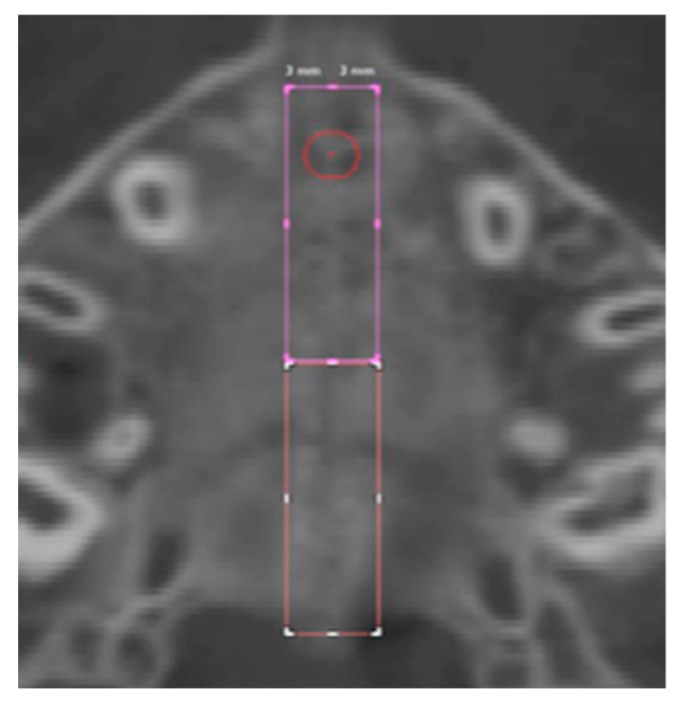
Rectangular-shaped ROIs.

**Table 1 medicina-56-00112-t001:** Palatal landmarks. A re-slice of the head was performed on the basis of the described landmarks in order to have the Left Palatal Foramen Point (LPFP) and the Right Palatal Foramen Point (RPFP) lying on the same axial scan, the Anterior Nasal Spine (ANS) and the Posterior Nasal Spine (PNS) on the same axial scan, and the ANS and the PNS on the same sagittal scan.

Landmark	Definition
Posterior Nasal Spine (PNS)	the most posterior point of the posterior nasal spine
Anterior Nasal Spine (ANS)	the most anterior point of the anterior nasal spine
Right Palatal Foramen Point (RPFP)	the most posterior and external point of the right palatal foramen
Left Palatal Foramen Point (LPFP)	the most posterior and external point of the left palatal foramen

**Table 2 medicina-56-00112-t002:** The RME group. Data are shown as mean and standard deviation (SD) at the two timepoints with the results of comparisons within the group, *p* < 0.05.

	T0		T1		
	Mean	SD	Mean	SD	Dependent *t*-Test
*AS ROI*	547.56	137.10	386.56	177.20	0.25
*PS ROI*	532.54	198.76	411.50	104.24	0.39
*AB ROI*	445.00	35.92	409.85	251.11	0.75
*PB ROI*	428.69	276.29	363.87	291.48	0.60
*ASD ROI*	519.58	123.61	391.03	59.87	0.18
*PSD ROI*	478.97	64.73	425.62	83.82	0.16

**Table 3 medicina-56-00112-t003:** The SME group. Mean and SD at the two timepoints and comparison within the group * *p* < 0.05.

	T0		T1		
	Mean	SD	Mean	SD	Dependent *t*-Test
*AS ROI*	390.79	141.37	295.98	162.55	0.12
*PS ROI*	560.05	162.27	387.81	146.91	0.01 *
*AB ROI*	366.61	216.71	322.91	244.77	0.28
*PB ROI*	357.67	204.98	344.66	263.92	0.85
*ASD ROI*	380.39	140.84	212.41	127.37	0.04 *
*PSD ROI*	326.91	163.92	193.70	142.60	0.04 *

**Table 4 medicina-56-00112-t004:** The RME versus the SME group. Mean and SD of the differences between the two timepoints with the results of the comparisons between groups.

	RME		SME		
	Mean	SD	Mean	SD	Independent *t*-Test
ASC	−161.00	267.12	−94.82	108.08	0.62
PSC	−121.04	282.87	−172.23	87.53	0.71
ASL	−35.15	230.97	−43.70	79.17	0.94
PSL	−64.82	255.12	−13.01	140.91	0.70
ASD	−128.55	174.70	−167.98	125.66	0.69
PSD	−53.35	68.35	−133.20	101.50	0.18

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
