# Peer review of "Midpalatal Suture Density Evaluation after Rapid and Slow Maxillary Expansion with a Low-Dose CT Protocol: A Retrospective Study"

_medicina, 2020, doi:10.3390/medicina56030112_

Round 1

Reviewer 1 Report

The study focuses on a topic of clinical interest in orthodontics, about possible advantages and disadvantages of slow maxillary expansion treatment vs the traditional rapid treatment.

Overall the manuscript is well written., but presents some criticisms to be addressed as follows:

In the introduction a more detailed description of the slow rate treatment would be of help for the readers. The sentence on page 2, lines 55-58 is not clear and mention the risk of accumulate large loads that need to be clarified.  Please provide some considerations about the clinical indications about the decision to apply a SME over the RME. Are the two treatments completely super-imposable and interchangeable? So which were the criteria used by the clinician to administer one treatment instead of the other?

Also, the aim of the work i.e. evaluation of the density in two different test groups of patients is too generic and no experimental hypothesis is stated.

In the Materials and Methods a general description of the patients is provided. However, it could be useful to prepare a table reporting the characteristics of the two groups in terms of N of males and females and of the presence of unilateral or bilateral posterior crossbite and the amount of maxillary transverse deficiency at the beginning of treatment, and the maxillary width and maxillo-mandibular relationship at the end of the retention period.

In the discussion the Author should comment the biological meaning of suture and bone remodelling and of suture separation vs bone deposition and specify the clinical risks if they exist.

Radiological figures are at too low resolution to allow the readers to appreciated the anatomical features.

Author Response

Dear Reviewer

thank you so much for the suggestions, we revised the manuscript following your indications. About the aim of the study we underlined that the evaluation of palatal suture density in two different test groups has been done in order to evaluate the skeletal effects related to these different protocols of palatal expansion. In the materials and methods section the characteristics of the two groups in terms of number of males and females is reported, while the type of crossbite and the amount of maxillary transverse deficiency has not been reported because the effects of RME and SME on transverse maxillary diameter has been yet published in a previous study (R. Martina, I. Cioffi, M. Farella, P. Leone, P. Manzo, G. Matarese, M. Portelli, R. Nucera, G. Cordasco “Transverse changes determined by rapid and slow maxillary expansion –a low-dose CT-based randomized controlled trial” Orthodontics and Craniofacial Research Vol. 15, Issue 3, August 2012;159-168), in this one instead the aim was different and was to evaluate the effects of RME and SME on suture density.

About the type of crossbite we have 27 monolateral crossbites and 3 bilateral; the bilateral crossbites has been treated 2 with RME and 1 with SME. In the discussion section we underlinde that a comparative evaluation of suture density between RME and SME is important to better comprehend the skeletal effects induced by these different protocols and to establish if a longer period of retention is needed to prevent relapses. If we do not adequately know the effects of slow and rapid palatal expansion on suture density, we can’t infact establish if retention timing is sufficient to avoid a skeletal relapse of palatal expansion. Radiological images has been revised and respect the DPI standard required from the journal.

Reviewer 2 Report

The authors provide a study about purely orthodontic treatment for children or adolescents without surgery. A maxillary expansion with a fast protocol (0.6 mm / d) is compared with a slow protocol (0.4 mm / week). Comparative times are before maxillary expansion (T0) and after 7 months of retention (T1). Points in the suture were also compared with points in the bone. With the fast protocol there are no density differences between T1 and T0, with the slow protocol there is a weak significance at some of the measuring points. There is no significant difference between fast and slow protocol. There are some critical points in this manuscript:

The manuscript showed plagiarism from other publication (see the attached pdf) and the presented figure 1 and table 1 is already published in : Cordasco, Giancarlo, et al. "Effects of orthopedic maxillary expansion on nasal cavity size in growing subjects: a low dose computer tomography clinical trial." International journal of pediatric otorhinolaryngology 76.11 (2012): 1547-1551. please rewrite the suspicious sentences and replace or remove the published photo.

Introduction:

Please mention in the introduction why there should be a difference between the procedures after the retention and what they indicate if present.

Material and methods:

Unfortunately I cannot really understand the ROIs (Regions of Interest). If there are already sample pictures, then you can clearly identify the places on a picture.

The ASD-ROI and PSD-ROI are probably the differences between RME and SME? Where exactly the missing signifcance ?.

Results:

Can you better work out the meaning of this result? Which procedure should be recommended based on the data?

Conclusion:

I would like to read the practical relevance based on the study results. What should I do now as an orthodontist, fast or slow?

Author Response

Dear Reviewer

thank you so much for the suggestions, we revised the manuscript following your indications. Plagiarism from other publication has been rewrited and published photos has been substitued with original ones.

We added a detailed description of the different region of interest, and of the method used for suture density evaluation.

About the meaning of the results we can only state that in the SME group seven months after tretament beginning, the midpalatal suture density is still not fully restored: this finding, that is different for the RME group, can be related also to the fact that treatment time duration was different in the two groups due to the expansion protocol, for this reason subjects performing SME underwent definitely shorter retention period if compared to RME group.

However no significant differences were detected between groups in suture density, suggesting similar rate of suture reorganization in spite of different retention period.

On the basis of the results of the present study is possible to state that, even if the dynamic of palatal expansion is different between rapid and slow protocols, there are not statistical significant differences on suture density. Greater level of evidence are necessary to better comprehend dental and skeletal effects induced by the different protocols of palatal expansions and to establish the more clinical effective and efficient protocol for palatal expansion.

Round 2

Reviewer 2 Report

Thank you for the revision, all requested changes were done

This manuscript is a resubmission of an earlier submission. The following is a list of the peer review reports and author responses from that submission.